# An Environmental Impact Calculator for 24-h Diet Recalls

**Thomas Bryan [1,\*], Andrea Hicks [2], Bruce Barrett [3] and Catherine Middlecamp [1]**

[1]  Nelson Institute for Environmental Studies, University of Wisconsin-Madison, Madison, WI 53706, USA; chmiddle@wisc.edu

[2]  Department of Civil and Environmental Engineering, University of Wisconsin-Madison, Madison, WI 53706, USA; hicks5@wisc.edu

[3]  Department of Family Medicine and Community Health, University of Wisconsin-Madison, Madison, WI 53706, USA; bruce.barrett@fammed.wisc.edu

\*  Correspondence: tbryan@wisc.edu

**Abstract:** The production of food is associated with significant environmental impact. In this paper, we describe the first assessment of the environmental impact of food consumption in the United States using individually reported dietary intake data from a nationally representative sample. Using individual-level dietary intake data from the National Health and Nutrition Examination Survey (NHANES) and applying median environmental impact factors compiled by Poore and Nemecek (2018), we estimate that the daily diet that a non-institutionalized U.S. civilian reports results in a mean of 3.92 $m^2$ (95% CI: 3.51–4.34) of land used, 2.26 kg (95% CI: 2.09–2.42) of $CO_2$e emitted, and 159 L (95% CI: 150–168) of freshwater withdrawn. The scope of all impacts is agricultural; transportation, storage, and preparation were not included. These results suggest that the calculator is ready for further development. This calculator can be used to estimate the environmental impact of individual diets in the 5100 studies (as of November 2018) registered with the Automated Self-Administered 24-h Dietary Assessment Tool, in addition to the last two decades of the nationally representative NHANES research.

**Keywords:** environmental impact; dietary assessment; automated self-administered dietary assessment tool (ASA24); 24-h dietary recall; National Health and Nutrition Examination Survey (NHANES)

---

## 1. Introduction

Food production occupies more than a third of the world's land surface and accounts for up to 30% of total anthropogenic greenhouse gas (GHG) emissions [1]. Agriculture accounts for 69% of freshwater withdrawals globally [2]. In the U.S. and the European Union, agriculture is the primary source of nutrient pollution in waterways, leading to eutrophication [3]. Food production is largely driven by food consumption. However, assessing the environmental impact of individually reported diets has not been well studied.

The environmental impacts associated with several hundred food items have been published. A 2018 meta-analysis of 1530 published studies lists the land use, carbon footprint, acidifying emissions, eutrophication emissions, freshwater withdrawal, and stress-weighted freshwater withdrawal impacts for 43 food groups [4]. The authors claimed that those 43 food groups account for ~90% of global protein and calorie consumption. In addition, review articles summarized the GHG emissions of food items [5–10]. Nonetheless, the data reported in these studies are all for food items, not for the individual diets of real people.

The environmental impacts associated with hypothetical or average diets (e.g., for vegetarian or Mediterranean diets) also are well documented. The researchers who estimate the footprints of these hypothetical diets often combine the environmental impact data of foods with different dietary scenarios [11]. Using this method, the impacts of these dietary scenarios can be forecasted. In addition, this method can be used to predict the dietary scenarios needed to reach national environmental targets [12]. In 2015, Hallstrom, Carlsson-Kanyama, and Borjesson reviewed the recent literature of dietary analyses for environmental impact [13]. All studies were based on average or hypothetical diets. None were based on individual diets of real people.

The objective of this study was to produce an environmental footprint calculator for individual diets; that is, real diets of real people. Previous studies have estimated the environmental impacts of individual foods, food groups, archetypal diets, and hypothetical dietary changes of entire countries. However, prior to this study, no off-the-shelf software existed for calculating the environmental impact of individual diets.

Since 2018, two research teams have developed multiple-factor environmental impact databases and linked them to individual diets. The first team from the University of Michigan and from Tulane University created dataFIELD (database of Food Impacts on the Environment for Linking to Diets) and linked it to nationally representative individual self-selected U.S. diets [14]. However, dataFIELD only contains two environmental impact factors: carbon footprint and non-renewable cumulative energy demand. Researchers on this team linked the database to 1-day dietary recall data of U.S. non-institutionalized citizens ($n$ = 16,800) in the 2005–2010 National Health and Nutrition Examination Survey (NHANES). The dataFIELD was expected to be made public in January 2019 [15].

The second team from Europe created an environmental impact factor database known as SHARP-ID (environmentally Sustainable, Health, Affordable, Reliable, and Preferred diets-Indicators Database) [16]. The team linked it to nationally representative dietary data from two non-consecutive days of 24-h recall records from Denmark, Czech Republic, Italy, and France. SHARP-ID also contains two environmental impact factors: carbon footprint and land use.

This study produced a tool similar to SHARP-ID and dataFIELD. However, this tool is unique because it is compatible both with NHANES and with any study that uses ASA24 (Automated Self-Administered 24-h Dietary Assessment Tool), of which there are over 5100 studies registered to use ASA24 as of November 2018 [17]. This tool also is unique because it utilizes a multi-factor environmental impact factor database (Poore and Nemecek 2018) that contains seven impact factors. As a result, we named it the miDIET (MultI-factor Dietary Impact on the Environment Tool). It is intended for research use and requires further development. It is published here because of our commitment to open-access research and our belief that the required further development is best done publicly.

## 2. Methods

Building an environmental footprint calculator for individual diets requires two things. First, people need to recall and record their diets for a 24-h period using a digital platform that identifies foods eaten and estimates the masses of those foods. Secondly, researchers need to match the foods these people ate with environmental impact factors. Our calculator allows for the automation of the second process so that environmental footprint of thousands of individual diets can be estimated at once.

### 2.1. Dietary Recall Data

Individuals can report their diets in many ways, including food frequency questionnaires, dietary history interviews, dietary records, and 24-h diet recalls [18]. Regardless of the method of reporting, two properties are essential for the miDIET: food identification and portion size estimation. We developed our calculator to be compatible with the Automated Self-Administered 24-h Dietary Assessment Tool (ASA24®) and NHANES.

The ASA24 is a digital platform for collecting 24-h diet recalls, identifies foods, and estimates portion size. It was developed by research scientists at the National Cancer Institute and at the U.S. National Institutes of Health [19]. As of November 2018, researchers from more than 5100 studies have registered to use ASA24 [17]. The ASA24 utilizes publicly-available databases and has been validated as a tool for 24-h dietary recall. The ASA24 identifies and estimates portion size of foods based on user input in accordance with the United States Department of Agriculture Food Nutrient Database for Dietary Studies (USDA FNDDS) 2013–2014 [20]. The validity of ASA24 has been tested multiple times [21,22]. Collectively, the validity studies conclude that diets recalled via ASA24 match true intake or performs equivalently to interviewer-administered methods of diet recall data collection. ASA24 is available in Canada and Australia, but the miDIET would need to be modified before adoption to ensure accuracy and applicability.

The National Health and Nutrition Examination Survey (NHANES) is a program of studies administered by the National Center for Health Statistics within the Center for Disease Control and Prevention. Since the 1960s, NHANES has been the nation-wide system for assessing the health and nutritional status of the American people. Each year, approximately 5000 participants provide individual-level data regarding their specific health and disease conditions. Physical exams and laboratory assessments complement self-reported health status to provide a detailed and representative picture of health in the United States.

NHANES uses the USDA Automatic-Multiple Pass Method (AMPM) to collect 24-h dietary recalls. It is administered by an interviewer in the Mobile Exam Center where respondents participate in interview and physical examinations. The ASA24 is based on AMPM. Most importantly, both output the food codes are defined by the USDA FNDDS. This makes NHANES and studies that use ASA24 compatible with the miDIET.

We tested the miDIET on the NHANES 2015–2016 dietary data. All analyses accounted for the NHANES survey design parameters and sampling weights. To account for the complex survey design of NHANES, we used the sampling weights 'wtdrd1' from the NHANES dietary data and primary sampling units 'sdmvspu' and strata 'sdmvstra' from NHANES demographic to analyze the data. To ensure reproducibility, the Stata (version SE 16.0; College Station, TX, 77845, USA) statistical file and the Excel-based miDIET tool are included in the Supplementary Materials. The test provides evidence that the miDIET can be used to estimate the environmental impact of dietary intake of individual U.S. non-institutionalized citizens. The test followed the instructions outlined in the ReadMe sheet (Supplementary Materials).

*2.2. Environmental Impact Factors*

A 2018 meta-analysis of 1530 published studies lists the land use, carbon footprint, acidifying emissions, eutrophication emissions, freshwater withdrawal, and stress-weighted freshwater withdrawal impacts of 43 food groups [4]. Each impact factor is expressed as impact per mass of food (Table 1). The scope of these impact factors does not extend beyond production, so we could not include other supply chain steps or consumer-level impacts in the miDIET.

**Table 1.** Impact categories and respective units from Poore and Nemecek 2018.

| Impact Category | Unit (per kg of Food) |
| --- | --- |
| Land use | $m^2$ |
| Greenhouse gas emissions | kg $CO_2$eq, IPCC 2013 includes feedbacks |
| Greenhouse gas emissions | kg $CO_2$eq, IPCC 2007 |
| Acidifying emissions | g $SO_2$eq, CML2 Baseline |
| Eutrophication emissions | g $PO_4^{3-}$eq, CML2 Baseline |
| Freshwater withdrawal | L |
| Stress-weighted water withdrawal | L |

To match the 43 food groups with the 8536 entries of the USDA FNDDS, hand-coding each entry was necessary. The English language nomenclature for entries in the USDA FNDDS is a strict hierarchy: food name, sub-category1, sub-category2, . . . sub-categoryX (e.g., "Bagel, wheat, with raisins"). The coding scheme for the 43 food groups takes advantage of this hierarchical nomenclature. For example, environmental impact factors for beef were applied to all USDA FNDDS entries that began "Beef, . . . ". Other USDA FNDDS entries that contained the word "beef" were considered in the hand-coding process. The median environmental impact factors from Poore and Nemecek 2018 were applied to those entries with a single predominant ingredient as judged during hand-coding. The resulting coding scheme is in the CodingScheme sheet of the miDIET (Supplementary Materials).

Yield factors were necessary to translate some functional units of the impact factors. For example, the functional unit for coffee was 1 kg of ground roasted beans in Poore and Nemecek 2018. However, people recall drinking cups of coffee, not mass of beans. Therefore, a yield factor for coffee and other food products allowed for those food groups to be included in the miDIET. All yield factors are in the miDIET (Supplementary Materials).

With the hand-coding complete, the miDIET can match any single dietary recall record with its corresponding environmental impact using the IndexMatch function in Microsoft Excel. Pre-programmed algebraic expressions convert arithmetic converts per kilogram values of the impact factors into per serving values using the mass of food in the dietary recall record. This function is automatically applied across all food recalled in any dietary recall record using a macro-enabled Excel workbook, allowing for automated, high-throughput calculation of the environmental impact of individual diets.

## 3. Results

This study yielded two results: the miDIET itself and its estimations of the environmental impact of individual diets based on a nationally representative sample.

### 3.1. The Multi-Factor Dietary Impact on the Environment Tool (miDIET)

The miDIET is a relational database located in the "FNDDS_ECO" worksheet of the Supplementary Materials. It contains four types of data. Two are from the USDA FNDDS: descriptions of the 8536 different foods and beverages and their respective unique eight-digit numeric food codes (Columns A and B). The other two are from Poore and Nemecek 2018: food categories from their meta-analysis and the environmental impact data for those categories (Columns C-Q). Together, these four types of data allow for the FNDDS eight-digit food code to query the relational database and produce environmental impact data for any given food recorded by an individual. That is, the miDIET will output impact data if and only if there is a positive match for that food between the USDA FNDDS and Poore and Nemecek 2018 databases.

The details of producing a positive match are described earlier in the Methods section. Of the 8538 foods listed in the USDA FNDDS, 2866 (33.6%) were positively matched to the 43 food groups from Poore and Nemecek 2018. In a few cases, the diets of individuals in our study consisted entirely of foods that are positively matched. However, in most cases, individuals ate foods that were not matched with environmental impact data (Table 2).

Not only is the miDIET the first of its kind, it is also compatible with the ASA24. This tool can be used to estimate the environmental impact of individual diets in the 5100 studies registered with the ASA24, as of November 2018 [17].

The miDIET is also automated. For Day 1 of 2 of NHANES 2015–2016, 121,482 food records were logged by 8505 participants. The automation allows it to estimate the environmental impact of those foods in about 10 s on a standard desktop computer, sum the impacts for each participant in pivot tables, and visualize aggregate data in pivot charts.

**Table 2.** Descriptive parameters of Poore and Nemecek and NHANES databases and their relationships.

| Parameter | Result |
|---|---|
| Food groups in Poore and Nemecek 2018 | 43 |
| Foods listed in FNDDS | 8538 |
| Matches (#) between FNDDS and Poore and Nemecek 2018 | 2866 |
| Matches (%) between FNDDS and Poore and Nemecek 2018 | 33.6% |
| Foods (#) in NHANES 2015–2016 (Day 1 of 2) | 121,482 |
| Foods (#) in NHANES 2015–2016 (Day 1 of 2) matched with environmental impact data | 41,928 |
| Foods (%) in NHANES 2015–2016 (Day 1 of 2) matched with environmental impact data | 34.5% |
| Calories (%) in NHANES 2015–2016 (Day 1 of 2) matched with environmental impact data | 24% |

## 3.2. Environmental Impact Estimations

For this proof-of-concept test we used Day 1 NHANES 2015–2016 data. From this analysis, we estimate that the daily diet that a non-institutionalized U.S. civilian reports results in a mean of 3.92 m$^2$ (95% CI: 3.51–4.34) of land used, 2.26 kg (95% CI: 2.09–2.42)of $CO_2$e emitted, and 159 L (95% CI: 150–168) of freshwater withdrawn. Further descriptive statistics can be found in Table 3. The distributions of the diets (excluding outliers) are in Figure 1. All analyses accounted for the NHANES sampling weights and survey design parameters. Linearized standard errors were calculated per NHANES analysis guidelines.

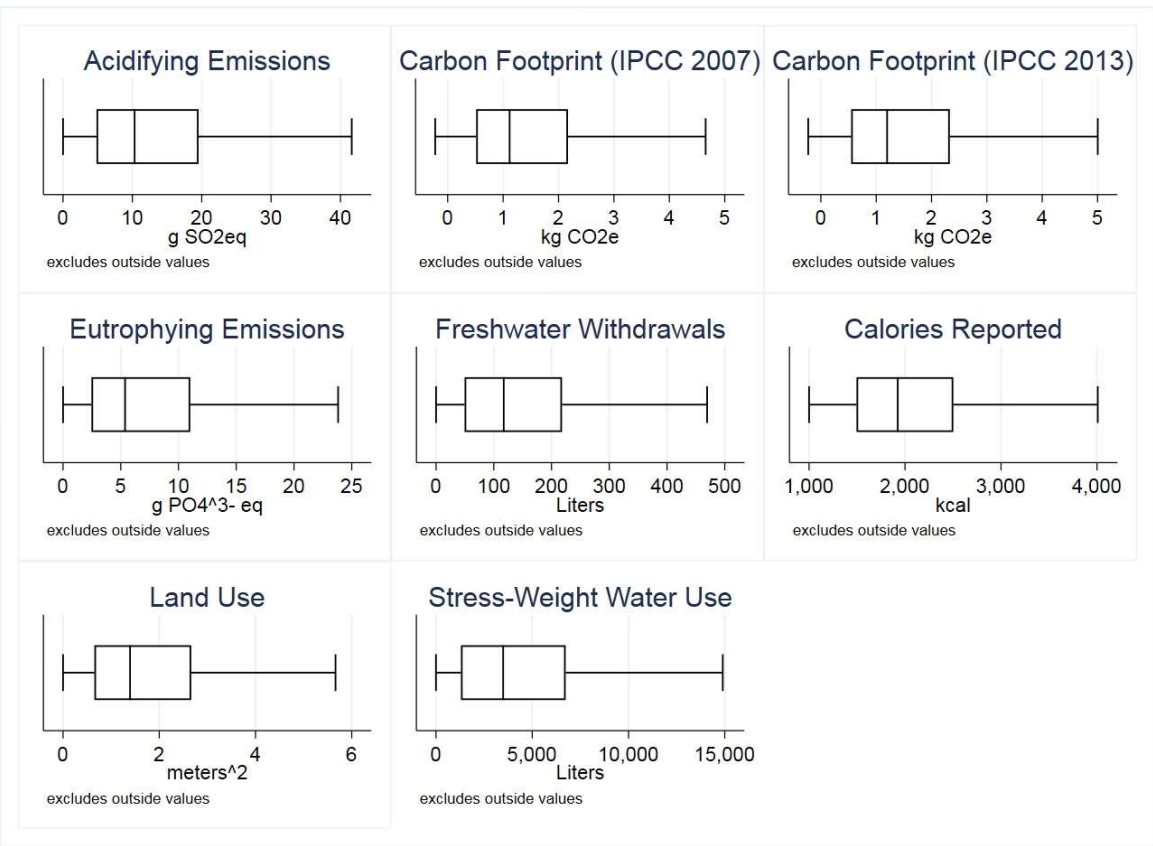

**Figure 1.** Modified box plots for environmental impact of NHANES 2015–2016 Day 1 diets. The middle line in the boxes represents the median. The upper and lower lines in the boxes represent upper and lower quartiles (Q3 and Q1, respectively). Whiskers end at upper and lower adjacent values (Q3 + 1.5IQR and Q1-1.5IQR, respectively). Values outside the adjacent values are excluded.

**Table 3.** Descriptive statistics of the miDIET outputs of Day 1 NHANES 2015–2016 dietary data.

|  | Mean | 95% Confidence Interval | Linearized Standard Error |
|---|---|---|---|
| kcal | 2020 | 1980–2060 | 17.6 |
| Land Use (m$^2$) | 3.92 | 3.51–4.34 | 0.195 |
| CF kg $CO_2$e (IPCC 2013) | 2.26 | 2.09–2.42 | 0.0777 |
| CF kg $CO_2$e (IPCC 2007) | 2.05 | 1.90–2.19 | 0.672 |
| Acidifying emissions g $SO_2$eq | 16.3 | 15.4–17.2 | 0.436 |
| Eutrophying emissions g $PO_4{}^3$q | 12.2 | 11.2–13.2 | 0.474 |
| Freshwater withdrawals (L) | 159 | 150–168 | 4.28 |
| Stress-weighted water use (L) | 5140 | 4780–5500 | 168 |

## 4. Discussion

The first iteration of any technology or methodology most likely have limitations. Some of these will be known, others not yet known. Here, we list four known limitations.

First, the scope of all estimated impacts is limited to agriculture. Other supply chain steps (i.e., transport, storage, and preparation) and consumer behaviors (i.e., packaging choices and food waste decisions) are not included, hence the results shown here should not be extrapolated beyond agriculture. Even so, production often is the most impactful stage in the life cycle of food [7,8,23].

Second, only 43 food groups with environmental impact data were included. As more environmental impact factors are published, the scope and hence accuracy of the tool will increase. Furthermore, the 43 food groups could be positively matched with only 33.6% of the foods in the USDA FNDDS. For example, beef burgundy (beef bourguignon, #27111200) is not matched with any food group from Poore and Nemecek 2018. The primary ingredient in beef burgundy is beef. A review of recipes could result in a yield factor that estimates just how much beef is in a serving of beef burgundy. Several hundred of these assumptions could be made to increase the relational capacity of the miDIET with the USDA FNDDS.

Third, the miDIET does not yet address the trove of nutritional information that can be assessed by the ASA24, as the dataFIELD developers have already done [24]. Sixty-five nutritional data points are collected for each food logged by an individual. Absolute environmental impacts without nutritional or caloric context are useful for relative and longitudinal analyses. However, including nutrients allows this calculator to become a valuable tool for cost benefit analyses. The environmental cost for nutritional benefit analyses includes calorie-by-impact, single nutrient-by-impact, and nutrient density-by-impact analyses. The EAT-Lancet Commission suggested that there is major potential for dietary changes to improve health and reduce the environmental impacts of food production [25]. This calculator can provide the individual-level dietary metrics to measure that potential when dietary changes are made.

Fourthly, the calculator relies on self-reported dietary intake records. While ASA24 is a validated tool for recording diets, it does not include food that is purchased but not eaten, i.e., food that is wasted by the consumer post-purchase. Furthermore, ASA24 does not account for food lost or wasted prior to sale, i.e., during processing, transport, or storage.

Despite these limitations, the results suggest that the calculator is ready for further development. As more food groups are studied for their environmental impact and added to the calculator, the performance of the calculator will only increase. Furthermore, more foods can be added to the calculator by making reasonable inferences regarding recipes, such as the earlier example of beef bourguignon. The United Nations Food and Agriculture Organization defines sustainable diets as those that are healthy, have a low environmental impact, are affordable, and culturally acceptable [26]. Using this definition, measuring and rating the sustainability of a diet will likely be a case-by-case endeavor. However, this calculator can begin measuring the health and environmental impact of a population's diet. As the first of its kind and with additional development, this calculator can become a high resolution, high throughput, and accurate environmental footprint calculator for individual diets.

## 5. Conclusions

The miDIET has demonstrated its value and is ready for further development. While it has much room for improvement, this calculator opens several avenues for future research. Thousands of such studies are already compatible with this new research tool. Even so, the authors envision integration of this calculator directly into ASA24. This would lower the barrier to entry for existing and future dietary researchers connecting the nutrition and environmental impact of individual diets.

**Supplementary Materials:** The following are available online at http://www.mdpi.com/2071-1050/11/23/6866/s1, The Microsoft Excel macro-enabled workbook is referred to as 'the calculator' in the research article. It contains a ReadMe worksheet, NHANES 2015–2016 Day 1 dietary intake data, a partial dataset from Poore and Nemecek 2018, a relational database that connects dietary intake data with environmental impact data, and the formulae and macros used for automated environmental impact estimation of individual diets. The statistical file used for data analysis is a STATA/SE 16 file. It contains NHANES participant numbers, total environmental impact data for each participant, and sampling weight, primary sampling unit, and strata demographic information for each participant.

**Author Contributions:** Conceptualization, T.B.; Data curation, T.B. and B.B.; Formal analysis, T.B.; Funding acquisition, B.B. and C.M.; Methodology, T.B. and A.H.; Project administration, B.B.; Software, T.B. and A.H.; Supervision, A.H., B.B. and C.M.; Visualization, T.B.; Writing—original draft, T.B.; Writing—review and editing, A.H., B.B. and C.M.

**Funding:** This research received no external funding. The APC was funded by Dr. Andrea Hick's start up funds from the University of Wisconsin-Madison Department of Civil and Environmental Engineering.

**Conflicts of Interest:** The authors declare no conflict of interest.

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
