# Peer review of "An Environmental Impact Calculator for 24-h Diet Recalls"

_sustainability, doi:10.3390/su11236866_

Round 1

Reviewer 1 Report

The reviewed manuscript was “An Environmental Impact Calculator for 24-Hour Diet Recalls”. The authors have developed a calculator for evaluating environmental effects of dietary intake. In the manuscripts, the authors describe the calculator and present some data calculated using the tool. The results concern environmental effects of diets of participants of NHANES in 2015-2017. This is a current and interesting topic. I have listed some major and some minor concerns.

My main concerns:

I am not convinced of accuracy of the data. Questions that arose are: The foods left out from the calculations comprise 86 % of the foods of the NHANES food record data and leave 76 % of total calories unmatched with environmental data. Do the authors assume that all the foods left out (because of missing environmental impact values) produce little environmental impact? What are the main food groups left out from the calculations that may possess high environmental impacts? How does the answer to the previous question affect the validity of this tool? How accurate do the authors think are the research results that were calculated from the NHANES data (keeping in mind the answers of the previous questions)? Are the results calculated from the NHANES data useful and how? I was expecting comparison of the amount of environmental impacts calculated from NHANES data with other studies or measures of environmental impacts of diets. Some description of the NHANES participants would be useful as results of them are presented.

Minor concerns:

I figured that this calculator is mainly for research purposes and not for consumer use. This could be clarified early in the manuscript. The authors state that ASA24 estimates portion sizes. I am unfamiliar with the tool and am left wondering what this actually means. How does it estimate them? More detailed description is necessary. In the discussion section, lines 163-164: Are limitations of ASA24 the only limiting factor? I mean, is such data of item-by-item life cycle data available? In the excel table, I do not understand the data on sheet “FNDDS_ECO”. Could the authors elaborate? What does NFS after some foods stand for?

Reviewer 2 Report

The authors present a novel classification of diet, based on environmental impact, as measured by kg of CO2 emitted, freshwater usage, and area of land used. This is an area of great interest and would be a good match to the readership of fields of sustainability and nutrition. There are some points of misinformation that should be corrected, and the review of the literature should be updated as there is other published work in this area. However, without the ability to classify a larger proportion of foods and beverages (34% match), the contribution of the calculator to the literature is questionable.

Abstract –

Line 14 - There are two other manuscripts (that I know of, there may be more) that describe the environmental impact of food consumption in the US. They only report on CO2 emission, and not freshwater usage or land use, but they should be acknowledged and integrated into the discussion.

Heller MC, Willits-Smith A, Meyer R, Keoleian GA, Rose D. Greenhouse gas emissions and energy use associated with production of individual self-selected US diets. Environ Res Lett. 2018 Apr;13(4).

Rose D, Heller MC, Willits-Smith AM, Meyer RJ.Carbon footprint of self-selected US diets: nutritional, demographic, and behavioral correlates. Am J Clin Nutr. 2019 Mar 1;109(3):526-534.

Methods

Lines 70-71 – do you have a citation for this number of studies registered to use ASA24?

Lines 78-79 – NO! NHANES does not use ASA24 to collect 24-hour recalls (24HR). NHANES uses the USDA’s Automatic-Multiple Pass Method (AMPM) to collect the 24HR. It is administered by an interviewer in the Mobile Exam Center. The ASA24 is based on AMPM, but there are striking and important differences. First and foremost, ASA24 it designed to be self-administered; however, it can be delivered by an interviewer. Second, ASA24 is totally auto-coded, allowing the foods and beverages consumed, in addition to the portions sizes, be immediately matched to food codes in the Food and Nutrient Database for Dietary Studies (FNDDS). AMPM must be collected by interviewers and is then coded by other individuals. I believe the proportion of foods that are auto-coded in AMPM is around 20-30%, but I don’t have a citation for that. Both tools are linked to FNDDS. This mistake needs to be corrected throughout the text.

Results

The match of only 34% of foods seems like the calculator isn’t ready for general use. I understand the complexity of assigning the environmental impact to mixed dishes (beef bourguignon), but there is a substantial gap between what has been match vs. what hasn’t been matched. Because of this, I’m unclear on the utility of this calculator.

As NHANES is publicly available data and uses FNDDS, it would be possible to use this calculator with ASA24. But it needs to be clear that NHANES does not use ASA24.

Conclusions

Please incorporate the other manuscripts referenced in this review in your manuscript.

Lines 187-188 – I’m not sure I agree that this calculator is ready for use with other datasets. Instead, I think substantial work needs to go into combination foods.

Reviewer 3 Report

The authors should be commended on their manuscript as it provides a unique contribution to the fast evolving area of research into sustainable diets. It is good to see some values on the environment impact of diets beyond hypothetical diets. Below are some minor comments.

It was acknowledged by the authors that the scope of this work was the agricultural impact of diets. If possible, the authors are encouraged to mention the approximate proportion of the overall environmental impact of diets/foods/food groups that transportation, processing, storage, preparation and household food waste are responsible for. This would help the reader interpret the findings.

The limitations associated with the proposed method and the assumptions made were outlined in the manuscript. However, it would be worth acknowledging that the values on the environmental impact of different foods are specific to the region/country and time of year they are produced. Therefore, the values used in this calculator may need to be modified if adopted by researchers in other continents that have access to the ASA24, such as Australia.

Limitations of the ASA24 were briefly mentioned in the manuscript. However, there are other dietary behaviours that play a key role in the environmental impact of one’s diet that are not assessed using this method, such as the use of individually packaged foods and beverages and the level of food processing.

I look forward to seeing how this method is further refined and applied by this research group in the future as I believe it will be of use to other researchers working in the area of sustainable diets. This manuscript would be of interest to the readers and I recommend it is published.

Round 2

Reviewer 2 Report

The authors have been very responses to suggestions to improve the manuscript.  Thank you.

I have a few other comments to consider:

It is unclear from your methods whether the complex survey design of NHANES was taken into account when calculating the estimates. Typically, the weight and survey design variables (PSU and strata) are used with NHANES to produce nationally representative data. While it is possible to use NHANES data without these, it is no longer nationally representative. It should be VERY clear and stated multiple times that 1)NHANES data was used as a test data set/casual data set, 2) weights and survey design factors were not applied, 3) results are NOT nationally representative.  (If this is the case).  If however, estimates were derived using these factors, then detailed information should be provided about what statistical package was used.  Additionally, standard errors should be presented, not standard deviations.

To confirm the ecocalc tab of the tool calculates the 7 environmental factors reported per food reported, correct?  So, these 7 columns could be exported into the appropriate statistical program so that the weights and survey design factors can be applied. This would increase the utility of the tool for use with NHANES data.  

Line 114 - NHANES is not limited to citizens of the US.  Suggest revising to 'respondents participate in..."

Line 173 Delete 'One such study is the latest".  As does not use ASA24 and it not reflected in the counts of studies registered with ASA24.

Line 174 Participants in the study recorded their diets for two days using USDA's AMPM (you currently have ASA24).

Round 3

Reviewer 2 Report

Again, the authors have been very responsive to comments. Thank you.

I have just one minor comment and I don’t feel I need to see the manuscript again. I am happy for it to go forward for publication.

Given that NHANES is a publicly available dataset, the analysis should be reproducible. To insure reproducibility, very clear and detailed methods are required. I would suggest looking at recently published manuscripts by authors at the National Center for Health Statistics, as this is the institute responsible for the NHANES data.

The text below was taken from a recent manuscript by a team of NCHS authors (Aoki Y, Brody DJ. “WIC Participation and Blood Lead Levels among Children 1-5 Years: 2007-2014.” Environ Health Perspect. 2018 Jun 29;126(6):067011. doi: 10.1289/EHP2384. eCollection 2018 Jun. PMID: 29961657). It is a good model for the language and types of information that should be included. The inclusion of code for STATA is a nice addition, but not entirely necessary. Apologies if a previous comment led you to believe I required that level of detail. Simply stating the statistical packaged used and that the complex sample design with the particular weight, is sufficient.

“Variances of estimates were calculated using Taylor series linearization per NHANES analysis guidelines, incorporating the complex sample design. We used prevalence ratios primarily as measures of association. Wald tests for testing group differences were performed on log(prevalence ratio), as was construction of confidence intervals (CIs) with back-transformation. Stata (version 13.0; College Station, TX) was used for all analyses.”
